



# Optimal Estimation Method Retrievals of Stratospheric Ozone Profiles from a DIAL Lidar

Ghazal Farhani[1], Robert J. Sica[1], Sophie Godin-Beekmann[2], and Alexander Haefele[3]

[1]Department of Physics and Astronomy, The University of Western Ontario, London, Canada
[2]Observatoire de Versailles Saint-Quentin-en-Yvelines, Guyancourt, France
[3]Federal Office of Meteorology and Climatology MeteoSwiss, Payerne, Switzerland

**Correspondence:** sica@uwo.ca

**Abstract.** This paper provides a detailed description of the first principle Optimal Estimation Method (OEM) which is applied to ozone retrieval analysis using Differential Absorption Lidar (DIAL) measurements. The air density, detector dead times, background coefficients, and lidar constants are simultaneously retrieved along with ozone density profiles. Using an averaging kernel, the OEM provides the vertical resolution of the retrieval as a function of altitude. A maximum acceptable height
at which the *a priori* has a small contribution to the retrieval is calculated for each profile as well. Moreover, a complete uncertainty budget including both systematic and statistical uncertainties is given for each individual retrieved profile. Long term stratospheric DIAL ozone measurements have been carried out at the Observatoire de Haute-Provence (OHP) since 1985. The OEM is applied to 3 nights of measurements at OHP during an intensive ozone campaign in July 2017 where coincident lidar-ozonesonde measurements are available. The retrieved ozone density profiles are in good agreement with both traditional
analysis and the ozonesonde measurements. For the three nights of measurements, below 15 km the difference between the OEM and the sonde profiles is less than 25%, at altitudes between 15 km to 25 km the difference is less than 10%, and the OEM can successfully catch many variations of ozone which are detected in the sonde profiles due to its ability to adjust its vertical resolution as the signal varies. Above 25 km the difference between the OEM and the sonde profiles does not exceed 20%.

## 1 Introduction

Stratospheric ozone plays a critical role, allowing life to thrive on Earth by absorbing the ultraviolet (UV) radiation emitted by the Sun. Moreover, the temperature structure in the stratosphere is determined by the absorption of UV radiation by ozone, which is followed by the exothermic recombination of $O_2$ and $O$. Thus, ozone is the main driver in defining the atmosphere's temperature structure (Andrews et al., 1987). Due to the high concentration of ozone depleting substances (ODSs), significant
chemical depletion of the total ozone has been detected globally since the mid 1980s (WMO, 1999). The ozone depletion is most pronounced in Antarctica, where the stratosphere is characterized by the presence of a strong polar vortex from May to November which isolates the air from ozone-rich air from the tropics. At the end of the Polar Night, when solar radiation returns, ODSs are activated yielding to the formation of species which cause ozone loss. As a result, most of the stratospheric ozone disappears (Farman et al., 1985; WMO, 2011, 2014). Substantial loss of stratospheric ozone in the vertical range of





14 km to 20 km was reported at times in the Arctic as well (Manney et al., 2011). Under the Montreal protocol, the emission and thus, abundance of anthropogenic ODSs in the troposphere has been decreased from its peak in 1994 by approximately 10% (WMO, 2014). Recently, the first signs of stratospheric ozone recovery over Antarctica were observed (Solomon et al., 2016). However, for non-polar regions, since 2000, no significant positive trend is detected (WMO, 2014).

Trends in ozone are in the order of a few percent per decade, e.g. in the upper stratosphere around +1% to +3% per decade (Harris et al., 2015). Although the trends in the total column ozone are insignificant, in the upper stratosphere (around 40 km) the ozone level has significantly increased (Harris et al., 2015). This increase does not indicate that ozone in the whole strato­sphere is increasing. In contrast, many studies have suggested that, at mid-latitudes and tropical latitudes, the ozone content in the lower stratosphere has continued to decrease (Ball et al., 2018). Thus, it is important to take ozone measurements with an

instrument with high spatial and temporal resolution to detect these changes.

DIAL (Differential Absorption Lidar) is a ground-based instrument which can measure ozone vertical distribution with high temporal and high spatial resolution. The technique also offers the advantage of making self-calibrated measurements. Details of the DIAL technique can be found elsewhere (Schotland, 1974). The traditional analysis of DIAL ozone measurements was presented by McDermid et al. (1990) and Godin-Beekmann et al. (2003). Recently Leblanc et al. (2016b) has presented a

detailed review of the method with full assessment of the random and systematic uncertainties. In this method, both statistical and systematic uncertainties are calculated. Moreover, count profiles from multi-channel systems must be merged to generate a single profile from multiple channels.

To determine a single ozone profile, the Optimal Estimation method (OEM) uses photocounts from multiple channels, without merging or applying corrections. Recently, the OEM has been implemented to lidar measurements to retrieve aerosol

backscatter profiles, Rayleigh temperature, and water vapour mixing ratio (Povey et al., 2014; Sica and Haefele, 2015, 2016). Here, we are applying the first principle of OEM to retrieve stratospheric ozone profiles from measurements at the Observatoire de Haute Provence (OHP) located in France. Ozone profiles are retrieved from raw (Level 0) measurements of four digital channels, two high altitude and two low altitude. Moreover, in this method, no pre- or post- filtering of retrievals is needed. The OEM provides a quantitative value for the maximum height of the retrieval. The uncertainty budget including both random

and systematic uncertainties is calculated on a profile-by-profile basis. This paper introduces a first principle OEM retrieval for stratospheric ozone density from DIAL measurements. In Section 2, the traditional analysis of ozone retrievals is discussed in details and is compared with the OEM algorithm. In Section 3, the approach to implement the OEM to the OHP lidar measurements is discussed in details. In Section 4, the OEM is applied to the night of 26 July 2017, and the result is compared with both ozonesonde measurements and the traditional analysis. The averaging kernel, vertical resolution of the retrieval and

systematic and statistical uncertainties of the retrieval are discussed as well. Moreover, the OEM results for two other nights are shown and are compared with the traditional analysis. Section 5 is the summary and our future work plans.





## 2 Methodology

### 2.1 The traditional DIAL method to determine ozone number density

In the DIAL technique, two laser beams at different wavelengths are simultaneously transmitted to the atmosphere. The spectral range for the laser beams is chosen in the UV range where one of the wavelengths is highly absorbed by ozone, and is called

the "on-line" wavelength. The other wavelength has a low absorption by ozone and is called the "off-line" wavelength. The measured backscattered photocounts, $N_{obs}(z,\lambda_i)$, for a laser pulse at wavelength $\lambda_i$, are given by the lidar equation (Fernald, 1984).

$$N_{obs}(z,\lambda_i) = \frac{C(\lambda_i)}{z^2}\beta(\lambda_i,z)\exp[-2\tau(z,\lambda_i)] + B(z) \tag{1}$$

where $C(\lambda_i)$ is the lidar constant, which contains the efficiency of the system, the telescope area, and the emitted number of

photons at each wavelength, $O(z)$ is the overlap function of the lidar, $\beta(z,\lambda_i)$ is the atmospheric backscattering coefficient, $\tau(z,\lambda_i)$ is the atmospheric optical depth, and $B(z)$ is the background photon counts. The atmospheric optical depth is given by:

$$\tau(z,\lambda) = \int\limits_{z_0}^{z}[\sigma_{O_3}(\lambda,T(z'))n_{O_3}(z') + \alpha(\lambda,z') + \sum_e \sigma_e(\lambda)n_e(z')]dz' \tag{2}$$

where $z_0$ is the altitude of the station, $\sigma_{O_3}(\lambda_i)$ is the ozone absorption cross section at the specific altitude and wavelength,

$T(z')$ is the atmospheric temperature, $n_{O_3}(z)$ is the ozone number density to be measured, $\alpha(\lambda,z)$ is the atmospheric extinction coefficient which includes both Rayleigh and Mie scattering extinction coefficients, and $\sum_e \sigma_e(\lambda)n_e(z)$ is the the extinction by other absorbers (like $SO_2$ and $NO_2$). In major volcanic eruptions the abundance of $SO_2$ gas in the stratosphere can significantly perturb the ozone retrievals. However, $SO_2$ only stays in the stratosphere for 30 to 40 days (Heath et al., 1983). In general, the amount of $SO_2$ mixing ratio in the stratosphere is negligible. The differential absorption cross section of $NO_2$ in the specified

spectrum is on the order of $3 \times 10^{-19}$ $cm^2$, thus considering the effect of $NO_2$ in the ozone retrievals is not essential, and the third term of Eq.2 is negligible (Godin-Beekmann et al., 2003).

For many lidar systems, at count rates below about 1 MHz, the relation between the true counts and the observed signal is linear. However, for the higher counts, the detector's response may not be linear. This relation for the non-paralyzable detectors is:

$$N_{obs} = \frac{N_{true}}{1 + \gamma N_{true}} \tag{3}$$

and for the paralyzable ones is:

$$N_{obs} = N_{true}\exp(-\gamma N_{true}) \tag{4}$$



where $N_{obs}$ is the observed counts, $N_{true}$ is the true counts, and $\gamma$ is the dead time. In the traditional method, the lidar measurements should be corrected for the effect of dead time. If the value of the dead time is not known, an empirical fit can be used to estimate the dead time value (Donovan et al., 1995). It is also well-known that for high intensity systems the output of the photomultiplier tube (PMT) can show an excess of counts some time after the signal intensity is maximum, a "tail" which is called signal-induced noise (SIN) (Hunt and Poultney, 1975). In fact, SIN is the residual signal originating from high signal intensities at low altitudes. It adds up with the background signal and is visible at altitudes where the signal-to-noise ratio (SNR) is very small (Iikura et al., 1987). Using a mechanical chopper to block high intensity light from approaching the detector is the most practical way to avoid SIN. It is important to consider the noise component from the upper altitude of lidar signals. In many lidars the background is a constant and the effect of SIN is not detected. If present, SIN is modeled using an exponential function of the form:

$$B(z) = e \exp(-fz) + g \tag{5}$$

where the fitting coefficients e, f, and g are analytically determined (Iikura et al., 1987). The SIN is more pronounced for the "on-line" wavelength, and for most nights its effect on the "off-line" wavelength is negligible, hence a constant background can be used.

## 2.2 Ozone Density Retrievals

In the traditional method, the derivative of the ratio between the "on-line" and "off-line" signals is calculated. The ozone number density can be retrieved as follows:

$$n_{o3(z)} = \frac{-1}{2\Delta\delta_{o3}(z)} \frac{d}{dz} \ln\left(\frac{N(\lambda_1, z) - B_1(z)}{N(\lambda_2, z) - B_2(z)}\right) + \delta n_{o3}(z) \tag{6}$$

where $N(\lambda_1, z)$ and $N(\lambda_2, z)$ are, respectively, the "on-line" and "off-line" signals at altitude $z$, $B_1(z)$ and $B_2(z)$ are the background signals, and $\Delta\delta_{o3}(z)$ is the differential absorption cross section between the two wavelengths. $\delta n_{o3}(z)$ is a correction term for the effect of differential Rayleigh and Mie scattering, and the differential absorption by other absorbers. More details can be found in McDermid et al. (1990); Godin-Beekmann et al. (2003); Leblanc et al. (2016b).

In the traditional ozone retrieval algorithm, several corrections are applied to the raw (Level 0) counts to produce corrected photocounts. For high count rates, the dead time of the counting system is determined and a non-linearity correction is applied. Depending on the configuration of the lidar, channels with different gains may be merged ("glued") to produce a single ozone profile. Determining the optimized height to merge the channels is typically done empirically. In the DIAL technique, the rapid decrease of sensitivity to ozone in the upper stratosphere is another important consideration. Low-pass filters are used to reduce the noise of the signals. For an ideal low-pass filter, the transfer function of all frequencies between 0 and the cut-off frequency, $\nu_c$ is 1, and the transfer function from $\nu_c$ to 1 is 0, where the reduced frequency $\nu$ is defined as $\frac{f}{f_N}$ and $f_N$ is the Nyquist frequency. The final vertical resolution of the signal, $\Delta z_f$, varies by the order of filter, which depends on the cutoff frequency





and the initial vertical resolution $\Delta z_i$:

$$\Delta z_f = \nu_c \Delta z_i. \tag{7}$$

A detailed discussion on the digital filtering and the vertical resolution can be found in Godin et al. (1999) and Leblanc et al. (2016a).

In the lower stratosphere, the perturbations in the ozone profiles are well detected; however, depending on the number of points in the filter (order of filter), the perturbation can be largely attenuated and cause negative or positive biases. For higher altitudes, because of the lower SNR, the vertical resolution is decreased. Different numerical filters have been tested to optimize the ozone retrievals. In all these techniques, to overcome the SNR decrease, the number of coefficients in the filters are increased with altitude (Godin et al., 1999).

**2.3   Applying the optimal estimation method to ozone retrievals**

The OEM is an inverse method in which the Bayesian theorem is used to find the probability distribution function (PDF) of the state of interest. Let $\mathbf{x} = (x_1, x_2, ..., x_n)$ be the state vector, and $\mathbf{y} = (y_1, y_2, ..., y_n)$ be the vector of the measurements. The relation between the measurements and the state vector is:

$$\mathbf{y} = F(\mathbf{x}, \mathbf{b}) + \epsilon \tag{8}$$

where $F(\mathbf{x}, \mathbf{b})$ is called the forward model. The forward model describes our understanding of the physics of the measurements as well as the instrument's characteristics. Here, $\mathbf{b}$ is the model parameter vector which contains additional parameters needed in the forward model, and the noise in the measurements is the vector $\epsilon$. In lidar measurements, the photon counts follow a Poisson distribution. However, for the counts rate more than 10 to 20, the PDF of the corresponding error tends toward a Gaussian distribution. Therefore, using the Bayesian approach and assuming a Gaussian PDF for all quantities, for a given

measurement $\mathbf{y}$, the most likely state of $\mathbf{x}$ is found by minimizing the following cost function:

$$\mathbf{J}(x) = [\mathbf{y} - F(\widehat{\mathbf{x}}, \mathbf{b})]^T \mathbf{S_y^{-1}} [\mathbf{y} - F(\widehat{\mathbf{x}}, \mathbf{b})] + [\widehat{\mathbf{x}} - \mathbf{x_a}]^T \mathbf{S}_a^{-1} [\widehat{\mathbf{x}} - \mathbf{x_a}] \tag{9}$$

where $\mathbf{S}_y$ is the covariance matrix of the measurements, $\mathbf{x_a}$ is the *a priori* profile which is an initial guess for the state vector, and $\mathbf{S}_a$ is the associated *a priori* covariance matrix. Typically, the cost is normalized to the number of measurements, and a cost of around 1 indicates a good retrieval.

As the forward model is nonlinear, the Marquardt–Levenberg method is used to find the state vector. The optimized solution for the state vector $\mathbf{x}$ occurs when the following iteration converges:

$$\mathbf{x}_{i+1} = \mathbf{x}_i + [(1 + \gamma_i)\mathbf{S}_a^{-1} + \mathbf{K}_i^T \mathbf{S}_y \mathbf{K}_i^T]^{-1} \left( [\mathbf{K}_i^T \mathbf{S}_y^{-1}(\mathbf{y} - F(\mathbf{x}_i)] - \mathbf{S}_a^{-1}(\mathbf{x}_i - \mathbf{x}a) \right) \tag{10}$$

here, $\mathbf{K} = \frac{dF}{dx}$ is the Jacobian of the forward model, and $\gamma_i$ is a damping factor for the iteration. A comprehensive description on the application of the Marquardt-Levenburg method to OEM can be found in (Rodgers, 2000).





## 2.4 Ozone DIAL Forward Model

Our first-principle OEM retrieval uses the lidar equation as the forward model and the raw counts are the measurements. The lidar equation for the true counts is:

$$N_{true}(z, \lambda_{on}) = (\frac{C_{\lambda_{off}}}{z^2})\beta(z, \lambda_{on})\Gamma_{O_3}(\lambda_{on}, z)\Gamma_{atm}(\lambda_{on}, z) + B_{\lambda_{on}}(z)$$

$$N_{true}(z, \lambda_{off}) = (\frac{C_{\lambda_{off}}}{z^2})\beta(z, \lambda_{off})\Gamma_{O_3}(\lambda_{off}, z)\Gamma_{atm}(\lambda_{off}, z) + B_{\lambda_{off}(z)} \qquad (11)$$

where $\lambda_{on}$ and $\lambda_{off}$ represents the "on-line" and "off-line" channels, $\Gamma_{O_3}(\lambda_{on,off}, z)$ and $\Gamma_{atm}(\lambda_{on,off}, z)$ are respectively, the ozone and atmospheric transmissions in each wavelength, $C_{\lambda_{on}}$ and $C_{\lambda_{off}}$ are the lidar constants, and $B_{\lambda_{on}(z)}$ and $B_{\lambda_{off}(z)}$ are the background counts. For the stratospheric ozone measurements, in the altitude region of retrieval, the overlap is complete, and thus, we have not included it in our forward model. Depending on the characteristics of the data acquisition system, the true counts are related to the observed counts by either Eq. 3 or 4. In multi-channel systems, our forward model calculates the

"on-line" and "off-line" wavelengths for both high altitude and low altitude channels. The transmissions are defined as:

$$\Gamma_{O_3,atm}(\lambda_i, z) = e^{-2\tau_{O_3,atm}} \qquad (12)$$

where the optical depth $\tau_{O_3,atm}$ is previously defined in Eq. 2. Both atmospheric optical depth and atmospheric backscattering coefficients have contributions due to scattering from molecules and aerosols:

$$\tau_{atm} = \tau_{mol} + \tau_{aer} = \int_{z_0}^{z} [\sigma_R n_{air}(z) + \alpha(z)]dz \qquad (13)$$

$$\beta_{atm} = \beta_{aer}(z) + \beta_{aer}(z) \qquad (14)$$

where $\beta_{air}(z)$ and $\beta_{aer}(z)$ are the corresponding air and aerosol backscattering coefficients. The "on-line" and "off-line" coefficients are related through the following equation:

$$\beta_{aer}(\lambda_{off}) = \beta_{aer}(\lambda_{on})(\frac{\lambda_{off}}{\lambda_{on}})^{-a} \qquad (15)$$

where for aerosols the Ångstrom coefficient $a$ equals approximately 1, and for molecular scattering the Ångstrom coefficient $a$

equals 4. In this paper, we only considered the clean-night condition. Therefore, the aerosol contribution to the process is not included, but could be in the future.

Due to the presence of SIN in the "on-line" channel, the background is assumed to be a function of height in the form of Eq. 5, while due to a negligible presence of SIN in the "off-line" channel, a constant background is used. If necessary, it is possible and easy to assign any analytic function for the background in both channels. Therefore, if needed the background for the



"off-line" channel can be assumed as a function of height as well. Using the above forward model, the ozone and air density profiles, the background coefficients, the dead time and the lidar constants for the 4 channels are simultaneously retrieved. Other parameters in the forward model are treated as model parameters. Hence, they are fixed but considered as a source of uncertainty contributing to the retrieved quantities (see Table. 1).

## 3 Implementing the optimal estimation method retrieval

To find the optimize solution of Eq. 10, *a priori* profiles for ozone and air density, as well as *a priori* values for background counts, dead time, and lidar constants are needed. Furthermore, **b** model parameter values and the covariance matrix of the measurements, *a priori* profiles, and model parameters need to be calculated. A summary of steps needed to implement the OEM for our ozone retrievals is shown in Fig. 1. A detailed description of these steps is provided in this section.

The *a priori* ozone profile used for all retrievals is from an OHP ozone climatology. The climatology contains monthly-averaged ozone profile using the last 30 years of OHP DIAL and SAGE II satellite overpass measurements. The standard deviation to the $2\sigma$ level for this climatology is 50% below 25 km and 10% above 20 km altitude. Alternatively, we have used the U.S. Standard model (Krueger and Minzner, 1976) as *a priori* ozone profile, which yields similar results for our ozone retrievals.

In the traditional method, the ratio of the "on-line" to "off-line" channels is calculated. Thus, there is no need to assume an air density profile to retrieve ozone. However, in the correction term (Eq. 6), the air density profile is needed and an atmospheric model or a measurement is used. In the OEM, we are retrieving the air density as a state vector, and the Mass Spectrometer Incoherent Scatter Radar (MSIS) air density profile is used as *a priori* profile. The MSIS profiles are generally in good agreement with the ozonesonde measurements of air density. An uncertainty of 15% is assigned to the *a priori* of air density.

In the case of ozone and air density there is a vertical correlation between the elements of retrieval states. This corresponds to the off-diagonal elements of the *a priori* covariance matrix. Generally, it is difficult to quantify the vertical length of this correlation. We have used a correlation length ($l_s$) of 1000 m for ozone at altitudes below 18 km and the correlation length of 1400 m at higher altitudes. The air density has a correlation length of 1400 m for all regions. A tent function is used to model the decay of correlation (Eriksson et al., 2005).

For the "off-line" channel the mean of the counts above 80 km are taken as *a priori* backgrounds, and their variances divided by the number of bins in the selected altitude region is used as *a priori* uncertainties in the background counts. For the "on-line" channel, an exponential function in the form of Eqn. 5 is fitted to counts above 80 km. The coefficients of the function are the *a priori* values. Depending on how good the initial fit is, uncertainties are assigned to the *a priori* coefficients, but for most nights a 20% uncertainty is chosen.

Using the forward model, the *a priori* lidar constants for both channels were estimated and an initial standard deviation of 10% for both channels is assigned. In a range in which photon counting measurements are linear (or non-linearity is cor-





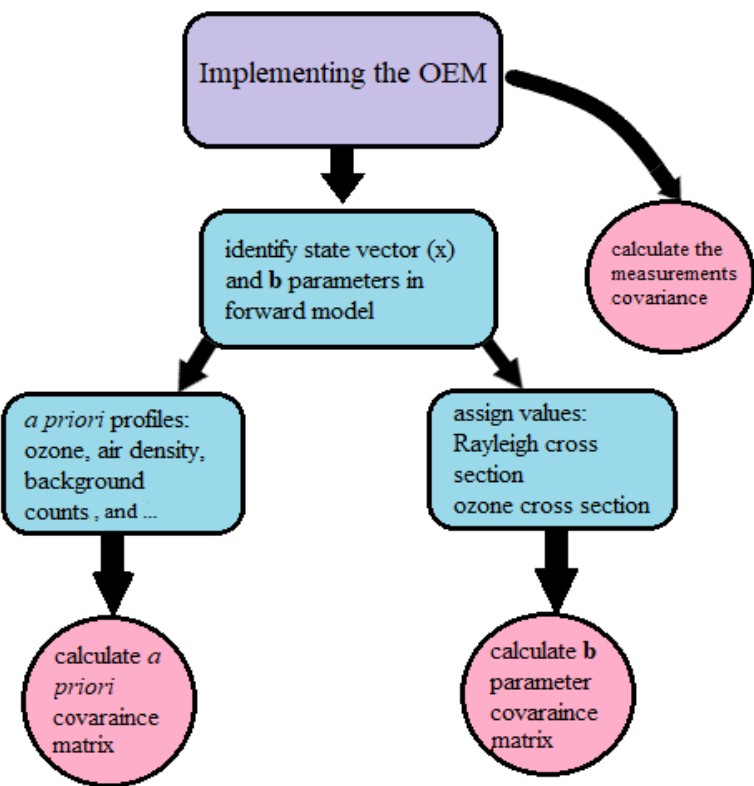

**Figure 1.** To implement the OEM, the *a priori* profiles for ozone and air density, background counts, dead time values, and lidar constants are needed. Moreover, **b** parameters should be identified and proper values for them should be calculated. The covariance matrices for *a priori* profiles, measurements, and **b** parameters need to be calculated as well.

rectable), Poisson statistics is applied. Thus, the measurement variances are the number of photons in each atmospheric layer located at altitude $\Delta z$, and there is no correlation between different layers (the off-diagonal elements of the matrix are zero).

The following quantities are calculated for the **b** parameters in the forward model. The Rayleigh extinction which is calculated using the Nicolet formula (Nicolet, 1984), and the temperature-dependent ozone absorption coefficients, as suggested

5    by (Orphal et al., 2016), are calculated based on the Brion–Daumont–Malicet (BDM) database (Malicet et al., 1995). Uncertainties of 0.3% and 2% (Leblanc et al., 2016a) are respectively assigned to the Rayleigh and ozone cross sections. The ozone absorption cross-section is a function of temperature. The BDM database provides values for 5 different temperatures; in order to find the ozone cross section for the whole region where ozone is retrieved, the temperature is interpolated. For the interpolation, the sonde temperature profiles are used at lower altitudes (up to the altitude at which sonde measurements are available)

10    ,and the MSIS temperature profiles are used for higher altitudes. Thus, the effect of temperature uncertainty on the ozone cross





section and the final retrievals needs to be calculated as well. An uncertainty of 19 K is assigned to sonde measurements of temperature, and an uncertainty of 35 K is used for the MSIS profiles. The covariance matrix of the **b** parameters will be used later to calculate the systematic uncertainty of the retrieved quantities.

Values and associated uncertainties of the *a priori* profiles for the parameters which we are retrieving, as well as the forward
model parameters which are considered as fixed parameters, (and thus, are not being retrieved) are summarized in Table. 1. As mentioned earlier, we are testing our model on a reasonably clear night condition from a high altitude site, therefore, we are assuming that the effects of aerosols are negligible. After calculating $\mathbf{S}_y$, $\mathbf{S}_a$, $\mathbf{S}_b$, $\mathbf{x}_a$, and **b** values, we used the Qpack software for our OEM retrieval. Qpack is a free Matlab package designed for forward and inverse modelling (Eriksson et al., 2005).

| Parameter | Value | Standard Deviation |
|---|---|---|
| Measurements | measured | Poisson statistics |
| Retrieved *a priori* values | | |
| Ozone density | OHP climatology | 50% to 10% |
| Air density | MSIS | 15% |
| Dead time | empirical fitting | 20% |
| Background ("off-line") | mean above 80 km | standard deviation above 80 km |
| Coefficients of SIN ("on-line") | empirical fitting above 80 km | 20% |
| Lidar constants | estimate from FM | 20% |
| Forward model parameters | | |
| Rayleigh-scatter cross section | Nicolet 1984 | 0.3% |
| Ozone absorption cross section | BDM 1986 | 2% |
| Temperature profile | sonde measurements | 19 K |
| Temperature profile | MSIS | 35 K |

**Table 1.** Values and associated uncertainties for the retrieved and forward model parameters.

## 4  Application of the OEM to measurements from the OHP stratospheric ozone lidar

OHP is located in the south of France at (44°N, 6°E, 650 m ASL). Long term stratospheric ozone DIAL measurements have been performed since 1985. In addition, the OHP lidar is part of the international Network for the Detection of Atmospheric Composition Change (NDACC). In the OHP DIAL system, the "on-line" wavelength is provided by a XeCl excimer laser emitting at 308 nm with an emission energy of 200 mJ and a repetition rate of 100 Hz. The "off-line" wavelength is generated from the third harmonic (355 nm) of a Continuum Nd:Yag laser, with an output energy of 40 mJ and the repetition rate of 50 Hz.
In the receiving end of the DIAL system, four similar F/3 mirrors of 0.53 m diameter collect the backscattered signals. The altitude steps of measurements is 150 m. The collected signal is separated to the Rayleigh signals at the transmitted wavelengths (308 nm and 353 nm), and the corresponding 1st Stokes wavelengths in the nitrogen Raman spectrum (332.8 nm and 386.7 nm).




Furthermore, to handle the high dynamic range of lidar signals in the whole altitude range, the Rayleigh signals are separated to the high and low gain channels. More details on the instrumentation can be found elsewhere (Godin-Beekmann et al., 2003).

The optical fibres transmit the receiving signals to the optical analysis device. The signals are detected by bialkali PMTs (Hamamatsu R2693P). The photon counting systems become nonlinear in the lowermost stratosphere. To correct for the saturation effect the following equation is used:

$$N_c = 1 + ((1-x)N_r - 1)\exp(-xN_r) \tag{16}$$

where $N_c$ is the observable counts, $N_r$ is the true counts, and $x$ is an adjustment parameter which equals the inverse of the maximum observed counts which is the definition of the dead time (Pelon and Mégie, 1982). To correct for the saturation, using Eq.16, the parameter $x$ is adjusted for each wavelength in order to get a best agreement between the slopes of high and low altitude signals. The altitude at which the two profiles are combined can vary from night to night (Godin-Beekmann et al., 2003). For the two wavelengths and two different altitude channels, the dead time can differ. Therefore, we are retrieving the dead times for each altitude and at each wavelength. A dead time value which corresponds to the $x$ parameter of each channel at each night is used as our *a priori*, and an uncertainty of $\pm 20\%$ is assigned to it.

Using the OEM, we retrieve the ozone density and air density profiles, as well as the dead time values for the four channels, the background counts for the "off-line" channel, and the SIN coefficients (three values) for the "on-line" channel. In total, we retrieved eight quantities along with the ozone density and air density profiles. The degree of freedom for our measurements, which is the trace of the averaging kernel, is $\approx 78$. Below we present the ozone retrieval for 26 July 2017 in detail. In order to show that the OEM is a robust method, the results for the nights of 14, and 20 of July are presented as well. In all these nights, ozonesonde balloons were coincidentally launched, thus the OEM is validated against both the traditional method and the sonde measurements.

### 4.1 Applying the OEM to OHP measurements on 26 July 2017

Figure. 2 shows the averaged counts over 4 hours of measurements for two different channels at "on-line" and "off-line" wavelengths on the night of 26 July 2017. The coincident ozonesonde is launched within one hour after the start of the measurement, and takes approximately 2 hours to reach 30 km. For each retrieval, the averaging kernel matrix is calculated. The averaging kernel is a diagnostic variable which describes how the retrieval sees changes in the real atmosphere. Therefore, it contains information on the sensitivity (area of the averaging kernel function) and on the smoothing (shape of the averaging kernel function) of the retrievals. Ideally the averaging kernel is a unity matrix preserving any change in the retrieved quantity from the *a priori* state. The area is defined as the vector product $\mathbf{Au}$ where $\mathbf{u}$ is a unit vector. When the retrieval comes solely from the measurements then the area equals 1, and at altitudes where the *a priori* profile is contributing to the retrievals the area decreases, where an area equal to 0 would mean nothing is being retrieved.

Figure. 3 shows the averaging kernels for the ozone density. The red line shows that the averaging kernel for ozone density equals 1 up to 42.7 km, thus below this altitude the retrieval is independent of the *a priori* profile. Ozone is a minor constituent




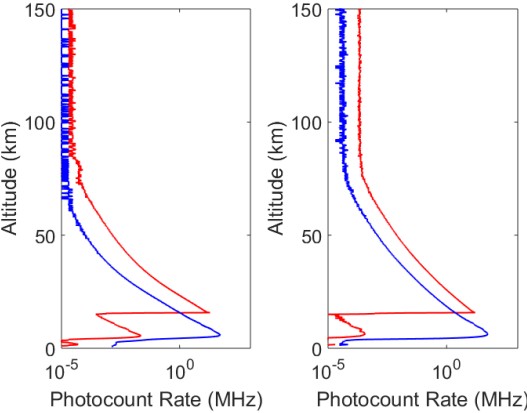

**Figure 2.** Average count rates for 5 hours of measurements on 26 July 2017. Left panel:"on-line" wavelength (blue curve, low altitude; red curve, high altitude). Right panel: "off-line" wavelength (blue curve, low altitude ; red curve, high altitude).

in the atmosphere; due to the poor SNR of signals at higher altitudes, the sensitivity of the averaging kernel decreases. Here, the retrieval falls back to the *a priori* values.

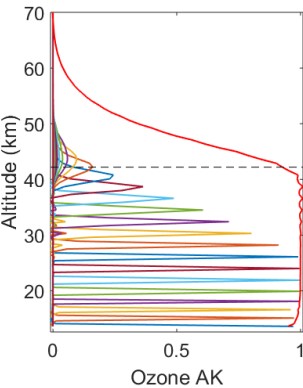

**Figure 3.** Averaging kernels for the ozone density for the measurements on 26 July 2017. The horizontal dashed line is a height below which the OEM retrievals is more than 80% due to the measurements. Above this horizontal cut-off as the SNR drops, the retrieval starts to fall back to the *a priori* profile. For clarity, the averaging kernels are only shown every 1500 m in altitude. The red line shows the summation of rows in the averaging kernel matrix at each altitude. The summation is of order unity below 42.7 km.

In a good retrieval, the difference between the forward model and the measurements, which is called the residual, should be within the uncertainty of the measurements. Figure. 4 shows the residual plots, which confirm that our forward model has correctly characterized the physics of the atmosphere and is capable of retrieving the quantity of interest.



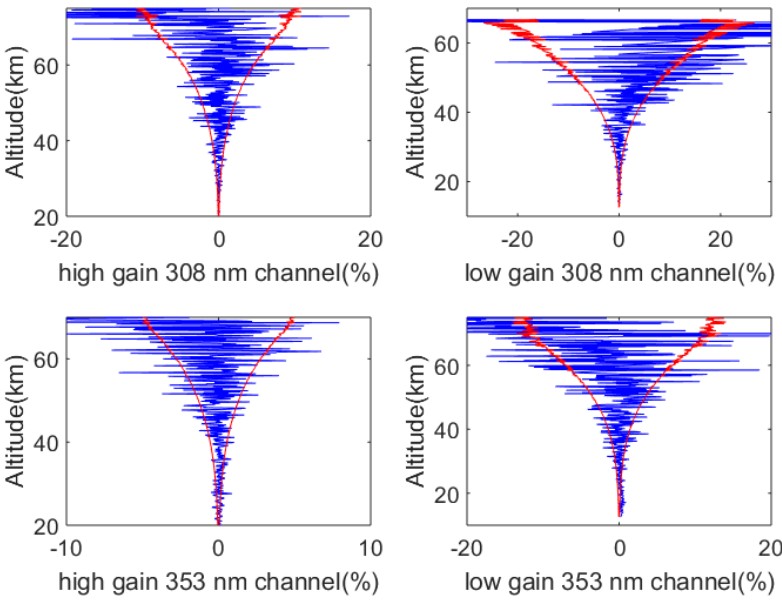

**Figure 4.** Residuals between the forward model and the measurements for the "on-line" and "off-line" channel (blue curves). The red line shows the uncertainty of the measurements.

The OEM retrieval grid starts at 500 m and increases to 700 m at 18 km. The full width half maximum of the averaging kernel at each height is defined as the vertical resolution of the retrieval. At lower altitudes, the averaging kernel is broad, and the retrieval resolution is close to the spacing of the retrieval grid (for this specific retrieval around 500 m). As shown in Fig. 5 (right panel), by increasing the altitude, the retrieval resolution decreases consequently, such that at 40 km the resolution is

2.8 km. Traditionally, the vertical resolution decreased by height as well. Figure.5 (left panel) shows the vertical resolution of the retrieval in both traditional methods and the OEM. At the first 2 km of retrieval the OEM provides a better retrieval resolution, however from 14.5 km to 17 km the traditional method has a better resolution. At around 17 km both methods show the same retrieval resolution; however, the traditional resolution decreases faster such that at 42.2 km the retrieval resolution is around 7 km. The trade-off between the retrieval resolution and the retrieval uncertainty should be considered when comparing

the methods.

Having a poorer vertical resolution leads to a better (that is, smaller) retrieval uncertainty. As shown in Fig. 5 (right panel), the statistical uncertainty of the retrievals for the traditional method is around 12% at 15 km (where the vertical resolution is 200 m and the low altitude Rayleigh channel is used) and it decreases to less than 1% at 25 km (where the vertical resolution is around 2 km and the high altitude Rayleigh channel is used). In contrast, the statistical uncertainty of retrieval in the OEM is

around 10% at 15 km (where the vertical resolution is 500 m) and decreases to 2.2% at 25 km (where the vertical resolution is 700 m).





To demonstrate the mentioned trade-off in the OEM, we increased the correlation length of the *a priori* from 1000 m to 1500 m in the lower altitudes (below 18 km) and from 1400 m to 5500 m in higher altitudes (above 18 km). As a result, the retrieval has a poorer vertical resolution and smaller retrieval uncertainties. Assuming a higher correlation length indicates that at each altitude, the retrieved ozone density is dependent on the ozone distribution above and below the indicated altitude, thus,

the retrieved ozone density looks smoother.

The vertical resolution and uncertainty for the traditional method as well as for the OEM with low and high correlation lengths are plotted in Fig.5.

In the traditional method, the relation between the final vertical resolution and the retrieval uncertainty is defined as follows:

$\quad \epsilon_s \propto (A \Delta z_f{}^3 P_0 t_a)^{-\frac{1}{2}}$ \hfill (17)

where $\epsilon_s$ is the retrieval uncertainty, $A$ is the area of the telescope, $\Delta z_f$ is the final vertical resolution, $P_0$ is the emitted power, and $t_a$ is the acquisition time (Godin et al., 1999). Assuming that the traditional method has the same vertical resolution as the OEM, using the above relation we can calculate the retrieval uncertainty which corresponds to the higher vertical resolution. Despite the difference in the vertical resolution values, at altitudes below 20 km, both the traditional method and the OEM

have similar uncertainties (the difference is less than 1%). At altitudes above 20 km, assuming that the traditional method has the same vertical resolution as OEM, the retrieval uncertainty in the traditional method is calculated. Figure. 6 shows the comparison between OEM uncertainty and the modified traditional uncertainty for altitudes above 20 km. As is shown in the figure, from 20 km to 35 km the difference between the uncertainties is insignificant (less than 1%) and above 35 km the difference grows to 4.5%.

Figure 7 shows our retrieved ozone density compared to the sonde measurements and the traditional retrieval. Consistent with Fig. 5 we have plotted the OEM retrievals for two different sets of correlation lengths. The ozonesonde measurements have better vertical resolutions compared to the DIAL measurements, albeit with larger random uncertainty. Also, the sonde profiles show more vertical structure of the ozone distribution. Compared to the traditional retrieval, the OEM can successfully catch many of these variations.

As shown in Fig. 7 (right panel) results of a comparison between the two methods indicates that for higher altitudes (above 25 km) the difference between the two retrievals is insignificant. However, for lower altitudes (between 15 km to 21 km) the difference between the two methods becomes significant. Moreover, as shown in the left panel of Fig. 7, on this particular night, the difference between the two methods below 15 km can be as large as 60% relative to the ozonesonde, with the OEM retrieval in better agreement with the sonde at most heights below 21 km. For higher altitudes the two methods agree well with

the sonde measurements.

To investigate the effect of *a priori* profiles on retrievals, the OHP climatology and the US standard model were used to retrieve ozone density (see Fig. 9). The OEM retrievals resulting from these two *a priori* profiles as well as the traditional retrieval are plotted in the left panel of Fig. 9. As shown in the right panel of this figure, below 35 km the difference between the two OEM retrievals is less than 0.5%. Above this altitude, the percentage difference between the two methods reaches 2.5%





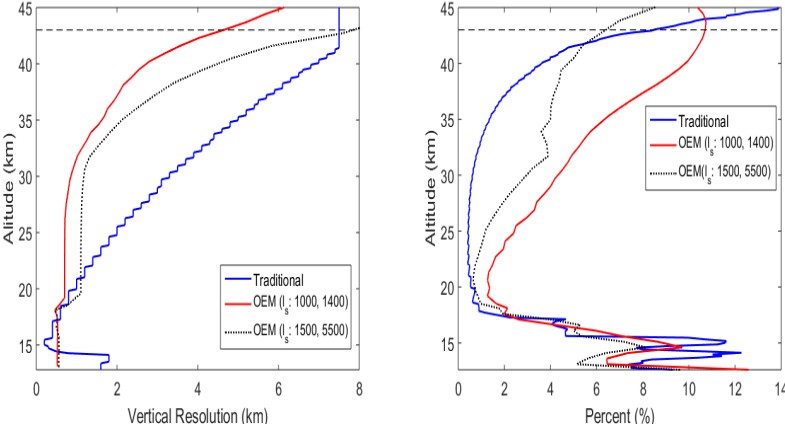

**Figure 5.** Right panel: The statistical uncertainty of the OEM with correlation lengths ($\ell_s$ = 1000 and 1400 m is plotted (red curve) against the statistical uncertainty of the OEM with correlation lengths $\ell_s$ = 1400 m and 5500 m (black dotted curve). Additionally, the uncertainty of retrieval in the traditional method (blue curve) is plotted. The retrieval uncertainties in the OEM and the traditional method can be compared. Left panel: The vertical resolution of the OEM with correlation lengths ($\ell_s$ = 1000 and 1400 m (red curve) is plotted against the vertical resolution of the OEM with correlation lengths $\ell_s$ = 1400 and 5500 m (black dotted curve). The vertical resolution of the traditional method is shown as well (blue curve). The horizontal dashed line indicated the maximum height at which the retrieval is independent from the *a priori*.

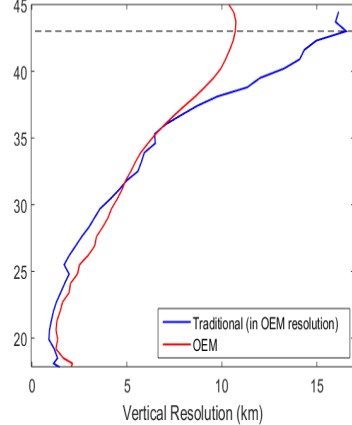

**Figure 6.** At height from 20 km to 40 km, the uncertainty of retrieval for the traditional method (assuming that it has a vertical resolution similar to the OEM vertical resolution) is plotted against the OEM retrieval uncertainty (blue curve: OEM; red curve: traditional). The horizontal dashed line indicated the maximum height at which the OEM retrieval is independent from the *a priori*.

which is much smaller than the retrieval uncertainty at altitudes above 35 km. Thus, the choice of *a priori* has a small effect on the retrievals.





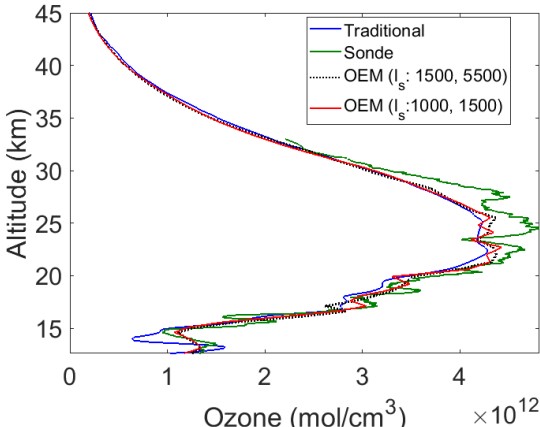

**Figure 7.** OEM ozone retrieval (red curve) from 20:07 UT to 00:15 UT on 26 July 2017 as well as the ozonesonde profile (green curve) and the traditional ozone retrieval (blue curve) are plotted. The dashed black line shows the OEM retrieval when the correlation length ($l_S$) became larger. The horizontal dashed line shows the cut-off below which the effect of the *a priori* ozone profile is small less tan 10%.

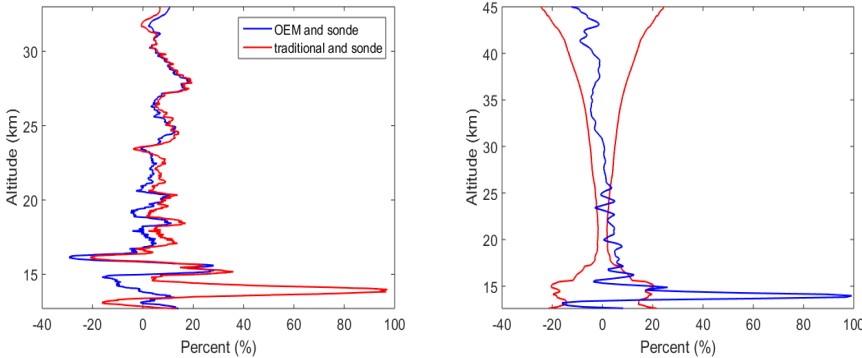

**Figure 8.** For the night of 26 July 2017. Left panel: The percentage difference between the OEM retrieval and the ozonesonde measurements in the form of: $\left(\frac{OEM-sonde}{sonde}*100\right)$ (blue curve); the percentage difference between the traditional retrieval and the ozonesonde measurements in the form of $\left(\frac{traditional-sonde}{sonde}*100\right)$ (red curve). Right panel: The percentage difference between the OEM retrieval and the traditional retrieval (blue curve); the summation of the statistical uncertainty of the traditional and OEM retrievals (red curve).

The OEM provides a complete systematic and statistical uncertainty budget. Fig. 10 shows the uncertainty of the ozone retrieval shown in Fig. 7. The forward model parameters, the Rayleigh cross sections, the ozone absorption cross section, and the temperature profiles assumed for the ozone cross section contribute to the systematic uncertainty of the retrieval. Below 20 km, these uncertainties are comparable with the statistical uncertainty; however, in the higher altitudes systematic uncertainties are less than 1%. The Rayleigh-scatter cross section uncertainty, at the bottom of the retrieval, is around 7% while at higher altitudes the uncertainty decreases to less than 1%. These values agree with the Rayleigh-scatter uncertainty of 8%





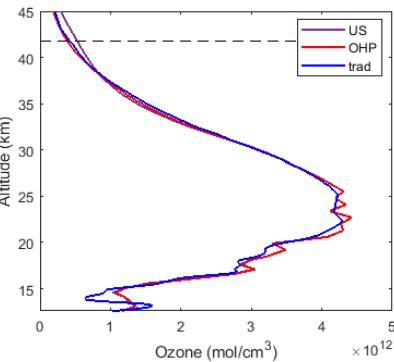
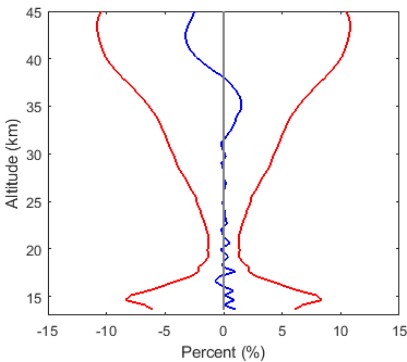

**Figure 9.** For the night of 26 July 2017. The left panel: the OEM retrieval using the US standard model as *a priori* profile (purple curve) and the OEM retrieval using the OHP climatology as *a priori* profile (red curve) are plotted. Furthermore, the traditional method retrieval (blue curve) is plotted, thus the OEM retrievals can be compared with each other and with the traditional retrieval. The right panel: Percentage difference between the OEM retrievals using the two different *a priori* profiles (blue curve) is plotted. This difference is with in the retrieval uncertainty. At higher altitudes (above 35 km), when the SNR drops, the difference between the two methods is less than 5%, which is smaller than the retrieval uncertainty at that height.

which is calculated in the Leblanc et al. (2016b) uncertainty budget. The ozone absorption cross section for 308 nm channel reached a maximum of 4% at the bottom of the retrieval, which is higher than the calculated uncertainty of 1% in Leblanc et al. (2016b). The uncertainty due to temperature is less than 0.05%. The uncertainty due to the ozone absorption cross section at 355 nm channel is negligible as well.

The ozone retrieval extends from 12 km to 70.2 km. The averaging kernel of the air density extends much higher, as the air density contributes in both back-scattering coefficients and the extinction coefficient terms in the forward model. Therefore, in air density retrievals, the maximum height of acceptable retrieval is 70.2 km. However, we show the retrievals below 42.7 km to be consistent with the ozone density retrievals. As shown in Fig. 11 (left panel), the relative air density profile is retrieved as well.

To validate our result, we used the nitrogen Raman spectrum at 386.7 nm. The "off-line" wavelength is transmitted to the atmosphere at 355 nm channel, and the corresponding Raman wavelength is received at 386.7 nm channel. The Raman channel is not sensitive to the aerosol contents of the atmosphere, and the wavelength is not absorbed by ozone ("off-line" Raman channel). Thus, the atmospheric back scattering and extinction terms are mostly determined by the air density. This makes the Raman "off-line" channel a good candidate for our validation. We can assume that $N(\lambda_{off}, z) \propto \frac{n_{air}}{z^2}$.

Using the above relation, the relative air density profile can be generated. The relative air density is scaled against the OEM retrieval of air density, and the percentage difference is calculated (Fig.11; center panel). As shown in the figure, the difference between the scaled relative air density generated from the Raman counts and the OEM relative air density is less than 10%. However, in higher altitudes (above 35 km) the difference can reach up to 50%. This difference is governed by the higher measurement noise in the Raman channel. This result provides confidence that the density retrieval is reasonable. The right





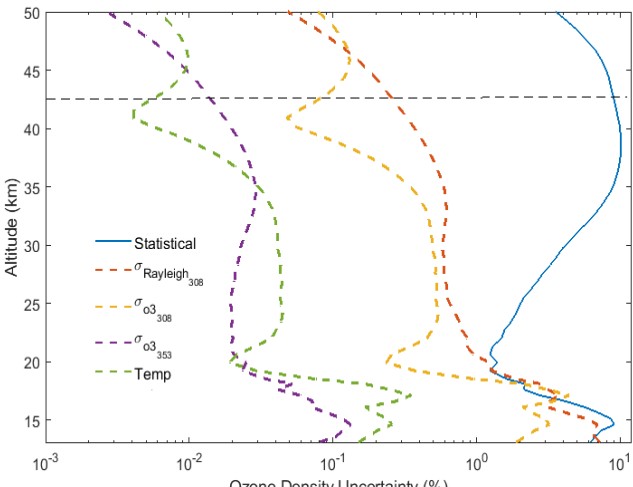

**Figure 10.** For the night of 26 July 2017. The statistical uncertainty of the OEM (blue), the Rayleigh-scatter cross section uncertainty at 308 nm (red), the ozone absorption cross section at 308 nm (orange), and the ozone absorption cross section for the 355 nm channel (purple). The horizontal dashed line shows the height below which the retrieval is independent of the *a priori* profile.

panel of Fig.11 shows the uncertainty of the relative air density retrieval. For the air density retrieval the statistical uncertainty is small (around 0.1% at the bottom of the retrieval). The Rayleigh-scatter cross section uncertainty is small as well and the ozone absorption cross section uncertainties are negligible.

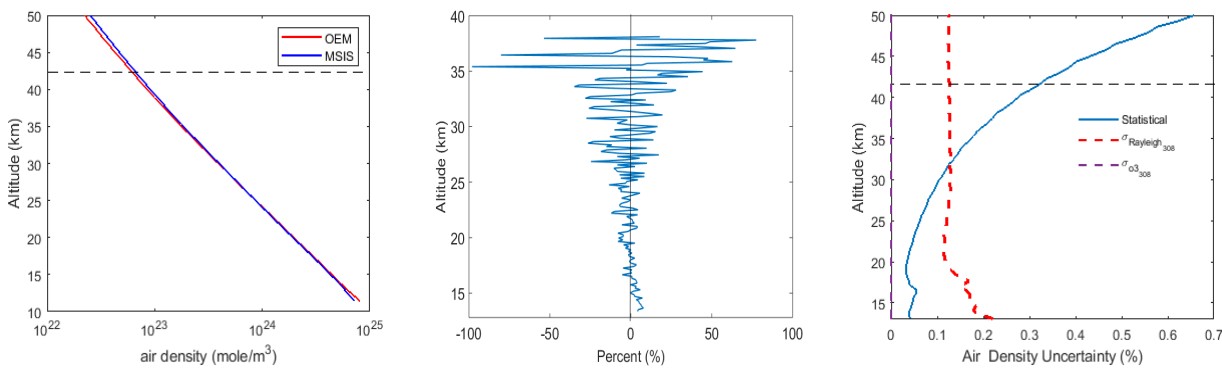

**Figure 11.** Left panel: The retrieved air density (blue line) is plotted against the *a priori* profile (red line). Mid panel: The percentage difference between the scaled relative air density generated from the Raman channel and the OEM air density retrievals. The difference is less than 10%. Right panel: The statistical uncertainty of the OEM retrieval of air density (blue), the Rayleigh-scatter cross section uncertainty for the 308 nm channel (red), and the ozone absorption cross section in both channels (purple).





| Dead time | OEM ( ns) | *a priori ( ns)* |
|-----------|-----------|------------------|
| "on-line" high-altitude | $2.78 \pm 0.55$ | 2.80 |
| "on-line" low-altitude | $5.05 \pm 0.92$ | 4.60 |
| "off-line" high-altitude | $4.60 \pm 0.92$ | 4.60 |
| "off-line" low-altitude | $2.56 \pm 0.51$ | 2.50 |

**Table 2.** Dead time values which were calculated for each channel on the night of 26 July 2017.

The OHP analysis employing the traditional method uses a different value of saturation correction for each wavelength. In our OEM code, we are retrieving 4 different dead times, each corresponding to one of the channels. For *a priori* values, we are using the provided $x$ value which is discussed earlier in this section. As shown in Table 2, the retrieved dead time values for 26 July 2017 are similar to the provided $x$ values. The only major difference is detected for the "on-line" low-altitude channel, where the $x$ value is 4.6 ns and the retrieved value is 5.05 ns.

## 4.2 Further examples of the OEM retrieval method

Using the OEM, the retrieved profiles for the nights of 14 July and 20 July are plotted against the sonde measurements as well as the traditional ozone retrievals (Fig.12). The night of 14 July 2017 includes 4.5 h of measurements. The retrieval extends from 9.6 km to 40.2 km. Above 16 km, the difference between the two traditional methods and the OEM retrieval is within the statistical uncertainty of the measurements. Below 16 km the difference is about 15% with the OEM retrieval closer to the sonde measurements (Fig. 13). For 20 July 2017 the retrieval is the result of 4 hours of measurements. The ozone retrieval extends from 11 km to 36.8 km. Our results indicate that the differences between the two methods are within the retrieval uncertainty. Thus, these two additional nights help to demonstrate that the OEM can produce ozone density profiles consistent with the traditional retrievals.

## 5 Conclusion

We have introduced a first-principle OEM retrieval for stratospheric ozone profiles applicable to stratospheric DIAL lidar measurements, and tested this method using measurements from the OHP stratospheric DIAL system. The discussion of the implementation of OEM for our retrievals is summarized below.

1. The forward model used in this study is capable of providing a robust estimate of the ozone profiles for clear nights.

2. Multiple measurements channels are used. The raw (uncorrected) photocounts are used for the retrieval, and no gluing process is needed. As a result, a single ozone profile consistent with all measurements is retrieved.

3. The OEM is applied to the OHP lidar measurements for three different nights in July 2017, all of which had coincident ozonesonde launches. Comparison with the radiosondes was good.





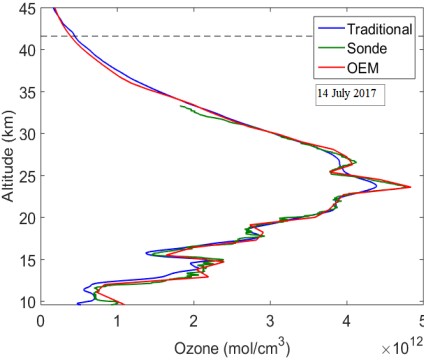 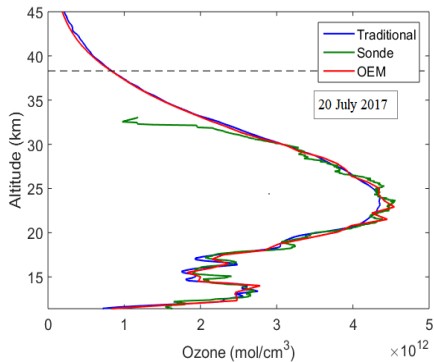

**Figure 12.** Left Panel: OEM ozone retrieval on the night of 14 July 2017 (red curve) compared to the ozonesonde profile (green curve) and the traditional ozone retrieval (blue curve). Right Panel: OEM ozone retrieval on the night of 20 July 2017 (red curve) compared to the ozonesonde profile (green curve) and the traditional ozone retrieval (blue curve). These cases demonstrate the high resolution of the OEM technique as evidenced by the excellent agreement around the ozone peak with the sonde measurement.

4. The OEM's averaging kernels allow the contribution of the *a priori* relative to the measurements to be accessed as a function of altitude, as well as allowing better comparison with other instrument.

5. The OEM and the traditional method are show good agreement, and for most heights their difference is small.

6. Increasing the correlation length in the retrieval allows the vertical resolution to be degraded and the statistical uncertainty decreased. Comparisons with the OEM retrievals at degraded resolution showed agreement to the traditional method to within the measurements statistical uncertainty.

7. The OEM provides a full uncertainty budget. Thus, using the OEM, for each individual retrieved profile both statistical and systematic uncertainties are calculated. The systematic uncertainties are compared with the uncertainty budget for the traditional method given by (Leblanc et al., 2016a) and are similar.

Currently we are working on a retrieval which can use measurements from both the OHP tropospheric and stratospheric lidars which will allow us to retrieve ozone profile from just above the boundary layer throughout the stratosphere. Also, we plan to include the Raman measurements into our forward model, allowing the retrieval of the ozone profiles in the presence of strong aerosol layers and thin clouds. Also, we are planning to apply our OEM retrieval to the last three decades of OHP measurements. Applying the OEM to the entire OHP lidar ozone profile database will provide an improved statistical evaluation of the differences between the traditional and the OEM methods, as well as allowing improved ozone estimates in the upper troposphere and lower stratospheric region.

*Acknowledgements.* We would like to thank Tom McElroy and Shayamila Mahagammulla Gamage for many interesting discussions about OEM, and Tom in particular for recognizing the potential for using OEM for lidar retrievals. We would like to thank Google engine which





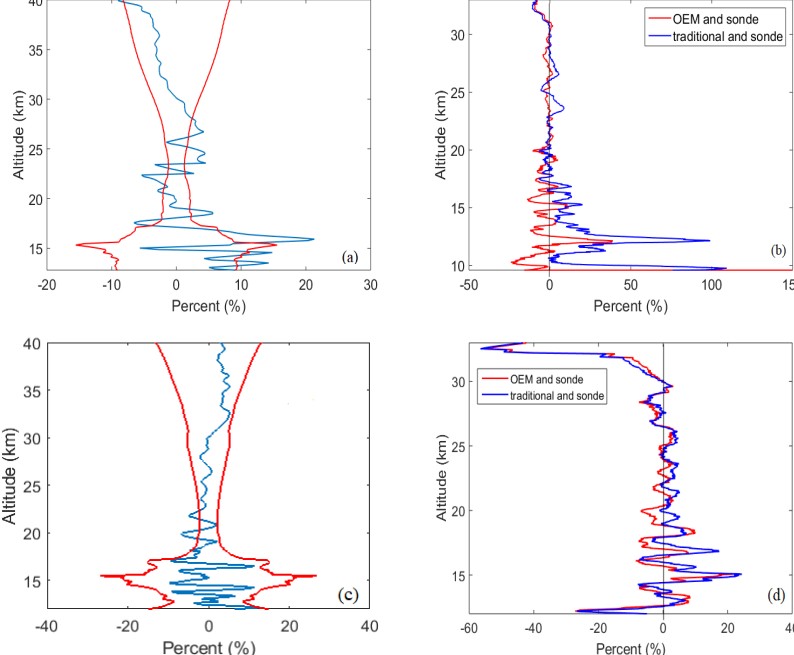

**Figure 13.** For the night of 14 July 2017, (a) the percentage difference between the traditional method and the OEM retrieval (blue curve) plotted within the envelope of the total statistical uncertainty of the two method (red curve). The agreement between the two lidar ozone determinations are within the statistical uncertainty above 17 km. (b): The red curve is the percentage difference between the OEM retrieval and sonde measurements. The blue curve is the percentage difference between the traditional method and sonde measurements. Figures (c) and (d) are the same format as (a) and (b) for the night of 20 July 2017.

selflessly helped us browse the world. We greatly appreciate the editing help from Patricia Sica, who patiently proof-read the manuscript. This project has been funded in part by the National Science and Engineering Research Council of Canada. The OHP lidar systems are funded by The National Center for Scientific Research. We would also like to thank the lidar operators of the station Gèrard Mègie at Haute-Provence Observatory.





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
