# Peer review of "Optimal Estimation Method Retrievals of Stratospheric Ozone Profiles from a DIAL Lidar"

_Atmospheric Measurement Techniques, 2018_

## Referee Comment (RC1) · Anonymous Referee #1 · 2 Nov 2018

Review for Optimal Estimation Method Retrievals of Stratospheric Ozone Profiles from a DIAL Lidar by Farhani et al., 2018

P2 L12, 'spatial' resolution may be an incorrect word choice

P2 L14, likely some older references to O3 DIAL (e.g. Megie)

P3 L5, 'relatively lower' absorption cross section as opposed to 'low'

P3L10, Overlap function $O(z)$ is introduced without the definition. $C = \ldots$ should be added.

P4, should add the aerosol component into eq. 2/6. It is fair to say that using ancillary data at OHP we have determined that aerosols were not a strong influence on these

retrievals. But, it should still be included in the OD calculation.

P7, L27, there will likely be fluorescence from the hydroxyl radical at 308nm between 80-100 km. Has this impacted the background subtraction model?

P12, L10 "The trade-off between the retrieval resolution and the retrieval uncertainty should be considered when comparing the methods." This is a subtle aspect of lidar analyses, is it possible to quantify this somehow. "Using the same VR scheme in the traditional and OEM results in a xx % increase in our statistical uncertainty assessment".

Fig 8 - OEM minus sonde appears to result in a negative value above 20 km. However, these curves are positive in this region. Is this actually Sonde minus OEM instead? Perhaps making the x-axis [-20,20] limits and a grid box would enhance this discussion as well. Also, at some point there is mention of Raman lower gain channels, are they also plotted on here? Is there differences there as well?

Figure 10, are there uncertainty associated with resolving times (dead times) as well? What about a final summed standard uncertainty on this plot? These uncertainty values of ~10% near 35km seem to be quite a bit larger than most traditional stratospheric DIAL measurements for an entire night. Is this evidence to modify the vertical resolution? This would be improved if you had the traditional uncertainty budget in a companion plot to show the differences.

Fig 12, it would be useful to have this figure with the different retrieval techniques using the same vertical resolution. It's not as straightforward to say one is picking out features and the other isn't if the VR is a factor of 3 different. Are the ozonesondes also being passed through the same low-pass filter for these comparisons?

Fig 13a Traditional DIAL background subtractions can be challenging to compare to 'truth'. Were there any other data sets (e.g. the NASA O3 Lidar), that would help bring closure to this discrepancy? Is this -10% bias in the top (<35km) could be attributed to

improper background subtraction?

---

## Referee Comment (RC2) · Anonymous Referee #2 · 16 Jan 2019

This is a valuable paper that is well-presented. I find no major issues and the paper should be published, with the minor corrections detailed below.

Minor comments:

Abstract: "first principle". We usually say "first principles".

Page 1, lines 20-24 & page 2, lines 1-4: This description of the ozone hole seems irrelevant to the paper.

Page 2, line 12: "The technique also offers the advantage of making self-calibrated measurements." Perhaps a word or two of clarification would be helpful here. Other differential absorption spectrometers need calibration.

[Figure]

Page 3, line 10: The overlap function is defined but does not appear in the equation.

Page 3, lines 19-21: Quoting the absolute value of the absorption cross section of NO2 tells the reader nothing unless the cross section of ozone is also given.

Page 3, lines 23 & 26: Is "paralyzable" a real word?

Page 7, line 12: "The standard deviation to the 2 sigma level for this climatology is 50% below 25 km and 10%..." From the context I think you mean the standard deviation (i.e. 1-sigma), but this sounds like a confidence interval. In Table 1 some of the other parameters are described as standard deviations.

Page 7, lines 19, 30 (and elsewhere): "uncertainty" is not defined. Do you mean one standard deviation (1-sigma)?

Page 7, line 23 (and elsewhere): "correlation length" is not defined.

Page 9, lines 1-2: An "uncertainty" of 19K for radiosondes sounds absurdly large. Uncertainties (1-sigma) for radiosondes are usually quoted in the range of 1K or less. I'm not familiar with MSIS uncertainties but 35K sounds pretty large. I can guess the temperature outside more accurately by simply looking at the calendar.

Table 1: I think you mean "Retrieval a priori values".

Page 10, lines 31-32: This description conflicts with that in the figure caption.

Page 13, lines 20-24: Why is the lidar biased low in Figure 7? It seems to miss quite a bit of ozone.

Page 13, line 25: Figure 8, not 7.

Captions, Figures 5 & 6: "...the maximum height at which the retrieval is independent from the a priori." is ambiguous (see Page 10, lines 31-32).

Caption, Figure 7: "...horizontal dashed line shows the cut-off below which the effect of the a priori ozone profile is small less than 10%." We now have a third definition of

this dashed line!

Figure 8 appears to be reversed (or the caption is wrong), as it shows the lidar higher than the sonde.

Page 16, line 5: "The ozone retrieval extends from 12 km to 70.2 km." How much of the upper part is a useful measurement?

---

## Author Comment (AC1) · 14 Mar 2019

**Response to the comments from the Referee #1: Optimal Estimation Method Retrievals of Stratospheric Ozone Profiles from a DIAL Lidar**

*Comment: P2 L12, 'spatial' resolution may be an incorrect word choice.*
Response: We changed it to vertical, so the sentence reads:

DIAL (Differential Absorption Lidar) is a ground-based instrument which can measure ozone vertical distribution with high temporal and high vertical resolution, especially in the lower stratosphere.

*Comment: P2 L14, likely some older references to O3 DIAL (e.g. Megie)*
Response: We added Megie et al. (1985) to the list of references.
The sentence reads:
The traditional analysis of DIAL ozone measurements was presented by Megie et al. (1985), McDermid et al. (1990) and Godin-Beekmann et al. (2003).

*Comment: P3 L5, 'relatively lower' absorption cross section as opposed to 'low'*
 Response: Done

*Comment: P3L10, Overlap function O(z) is introduced without the definition. C = ... should be added.*
 Response: The O(z) term is added to Eq. 1.

*Comment: P4, should add the aerosol component into eq. 2/6. It is fair to say that using ancillary data at OHP we have determined that aerosols were not a strong influence on these retrievals. But, it should still be included in the OD calculation.*

Response:  The aerosol term is embedded in the correction term. Note that the aerosol differential correction term includes a term linked to differential backscattering and differential extinction (as explained in Godin-Beekmann et al., 2003.).
Moreover, in page 3, line 15, we changed Rayleigh and Mie with molecular and particulate.
The sentence reads as:
$\alpha(\lambda,z)$ is the atmospheric extinction coefficient which includes both molecular and particulate scattering extinction coefficients

*Comment: P7, L27, there will likely be fluorescence from the hydroxyl radical at 308 nm between 80-100 km. Has this impacted the background subtraction model?*
 Response: This is an important effect to consider, but the OHP system has never observed an influence from OH fluorescence at 308 nm.

*Comment: P12, L10 "The trade-off between the retrieval resolution and the retrieval uncertainty should be considered when comparing the methods." This is a subtle aspect of lidar analyses, is*

*it possible to quantify this somehow. "Using the same VR scheme in the traditional and OEM results in a xx % increase in our statistical uncertainty assessment".*

Response: this is a good point, it is not clear in the current manuscript that we will go on to investigate this in detail. We propose changing the sentence to "... when comparing the methods and the reader is referred to the discussion below.

*Comment: Fig 8 - OEM minus sonde appears to result in a negative value above 20 km. However, these curves are positive in this region. Is this actually Sonde minus OEM instead? Perhaps making the x-axis [-20,20] limits and a grid box would enhance this discussion as well. Also, at some point there is mention of Raman lower gain channels, are they also plotted on here? Is there differences there as well?*

Response: Thanks for catching our mistake. Yes, it is (Sonde-OEM), we corrected this in our manuscript. And we changed the figure and add the following paragraph to explain the figure:

In order to account for the higher vertical resolution of the ozonesonde measurements, we use the OEM averaging kernels to ``degrade" (smooth) the sonde profile using:
\begin{equation}
x_{smoothed} = \mathbf{A} + ( \mathbf{I}_{n}- \mathbf{A}) \mathbf{x_{a}}
\end{equation}
where $\mathbf{I}_{n}$ is the unity matrix, and $x_{smoothed}$ is the smoothed sonde profile. Fig. \ref{fig:comparison} (left panel) shows the percentage difference between the smoothed sonde and the OEM (in blue) as well as the percentage difference between the smoothed sonde and the traditional profile (in red).

[Figure]

For the current retrieval, the Raman channels are not used. We plan in the future to include them. Our forward model is based on using the Rayleigh channels. However, we use the product of the ''off-line'' Raman channel counts and z^2 to generate a quantity proportional to the air density. The percentage difference between the retrieved air density and the above quantity is plotted in Fig. 11 (which became Fig 12 of the edited version).

*Comment: Figure 10, are there uncertainty associated with resolving times (dead times) as well? What about a final summed standard uncertainty on this plot? These uncertainty values of ~10% near 35 km seem to be quite a bit larger than most traditional stratospheric DIAL measurements for an entire night. Is this evidence to modify the vertical resolution? This would be improved if you had the traditional uncertainty budget in a companion plot to show the differences.*

Response: Table 2 in the manuscript addresses the first comment about the deadtime. We also added the traditional uncertainty budget figure, in the right panel of Fig.10 (Fig. 11 of the edited version). The traditional uncertainty budget is also shown below:

[Figure]

We have added the following paragraph to the manuscript on page 16:

The calculated OEM uncertainty can be compared with the traditional uncertainty budget. Fig. \ref{fig:uncertainty} (right panel) shows the uncertainty of the traditional ozone profile. The Rayleigh-scatter cross section uncertainty has a maximum value of 8\% at the bottom of the profile, while above 20\,km it becomes less than 1\%. This result is consistent with the uncertainty calculated by our OEM retrieval.

In the traditional analysis, for an isothermal atmosphere, the ozone absorption cross section uncertainty at 308\,nm is 3\%. The ozone absorption cross section uncertainty in our OEM retrieval is similar to \cite{LeblancThierry2016Part2}, whose Monte Carlo simulations allowed temperature to vary with height. In the traditional analysis, the background aerosol uncertainty is also calculated, which impacts the ozone profile by less than 1\% in the lower stratosphere. Aerosols are currently being added to the OEM forward model as a model parameter.
The statistical uncertainty of the traditional analysis at higher altitudes (above 25\,km) is smaller comparing to the OEM, which as explained earlier is the result of having a larger vertical resolution. However as shown in Fig.~\ref{fig:VR} (the black dotted lines), the OEM retrievals also have smaller statistical uncertainties if the vertical resolution increases. As discussed previously (Fig.~\ref{fig:VR-comparison}), for the traditional analysis using a similar vertical resolution to our OEM, the statistical uncertainty of the traditional method will be larger than for the OEM retrievals in the upper stratosphere, due to the regularization term in the OEM.

And, we added the following on page 5 explaining how we calculate the uncertainty budget:

The statistical uncertainty of the retrieved quantities and the model parameter uncertainties are calculated as follows:
\begin{equation}
\label{uncertainty_equ}
\begin{split}
\mathbf{S_{m}} = &\mathbf{G_{y}}\mathbf{S_{y}}\mathbf{G_{y}^{T}}\\
\mathbf{S_{f}} =
&\mathbf{G_{y}}\mathbf{K_{b}}\mathbf{S_{b}}\mathbf{G_{y}^{T}}\mathbf{K_{b}^{T}}\\
\end{split}
\end{equation}
where $\mathbf S_{m}$, $\mathbf S_{f}$, $\mathbf S_{b}$ are the covariances of the retrieval noise, the forward model parameter error, and the error covariance of the model parameters. The gain matrix, $\mathbf{G_{y}} = \frac{d\widehat {\mathbf{x}}}{d\mathbf{y}}$, gives the sensitivity of the retrieval to the measurements, while $\mathbf{K_{b}} = \frac{d {\mathbf{F}}}{d\mathbf{b}}$ is the Jacobian of the forward model with respect to $\mathbf{b}$.

*Comment: Fig 12, it would be useful to have this figure with the different retrieval techniques using the same vertical resolution. It's not a straight forward to say one is picking out features and the other isn't if the VR is a factor of 3 different. Are the ozonesondes also being passed through the same low-pass filter for these comparisons?*
Response: We propose adding a figure comparing the statistical uncertainty of the traditional ozone profile calculated with the same vertical resolution as the OEM ( Fig. 7 of the edited version and below):

[Figure]

Also on page 13:

The traditional ozone profile can be calculated at a similar vertical resolution to our OEM retrieval. The statistical uncertainty of the traditional analysis, using the same vertical resolution as our OEM, is shown in Fig.~\ref{fig:VR-comparison_2}. Below 30\,km both methods provide the same uncertainties, however, above this altitude the OEM uncertainty is smaller. The OEM's smaller statistical uncertainty at higher altitudes increases more slowly than for the traditional method due to the contribution of the \textit{a priori} profile, which adds additional information. However, in the OEM retrieval an increased contribution from the regularization term of the solution means the response function becomes less than 1. Below 30\,km the \textit{a priori} profile has a small contribution in the final retrieval (as the response function is $\sim 1$), but between 30\,km to 40\,km the \textit{a priori} profile has a greater contribution. Above 40\,km the response function decreases rapidly (Fig.~\ref{fig:AK}).

*Are the ozonesondes also being passed through the same low-pass filter for these comparisons?*

Response: As mentioned in an earlier comment, we changed all the comparison figures to make sure that the ozone sondes have the vertical resolution equivalent to the retrievals.

*Comment: Fig 13a Traditional DIAL background subtractions can be challenging to compare to 'truth'. Were there any other data sets (e.g. the NASA O3 Lidar), that would help bring closure to this discrepancy? Is this -10% bias in the top (<35km) could be attributed to improper background subtraction?*

Response: The background subtraction is not an issue at 35 km. For the NASA O3 lidar, the figure below representing both profiles on that day does not show a lower value at 35 km by the NASA lidar.
We would need to evaluate all OHP profiles by OEM method in order to evaluate biases between them. In fact the profiles of July 20 does not show the same discrepancy.

---

## Author Comment (AC2)

**Response to the Referee #2 comments on Optimal Estimation Method Retrievals of Stratospheric Ozone Profiles from a DIAL Lidar**

*Comment: Abstract: "first principle". We usually say "first principles".*
Response: fixed!

*Comment: Page 1, lines 20-24 & page 2, lines 1-4: This description of the ozone hole seems irrelevant to the paper.*
Response: We agree with you and have re-written this paragraph as follows.

Stratospheric ozone plays a critical role, allowing life to thrive on Earth by absorbing the ultraviolet (UV) radiation emitted by the Sun. Moreover, the temperature structure in the stratosphere is determined by the absorption of UV radiation by ozone, which is followed by the exothermic recombination of $\mathrm{O}_2$ and $\mathrm{O}$. Thus, ozone is the main driver in defining the atmosphere's temperature structure \citep{andrews1987middle}.
After observing a significant global depletion of stratospheric ozone \citep{farman1985large, WMO11,WMO14}, the Montreal protocol was established as an international treaty to control and to halt the release of ozone depleting substances (ODSs).  As a result, the abundance of anthropogenic ODSs in the troposphere has been decreased from its peak in 1994 by approximately 10\% \citep{WMO14}. Recently, the first signs of stratospheric ozone recovery over Antarctica was observed \citep{solomon2016emergence}. However, for non-polar regions since 2000, no significant positive trend has been detected \citep{WMO14}.

*Page 2, line 12: "The technique also offers the advantage of making self-calibrated measurements." Perhaps a word or two of clarification would be helpful here. Other differential absorption spectrometers need calibration.*
Response: We are adding the following to the paragraph.
DIAL (Differential Absorption Lidar) measures vertical distribution of ozone density with high temporal and vertical resolution. In the DIAL technique, two laser beams at different wavelengths are simultaneously transmitted to the atmosphere. The spectral range for the laser beams is chosen in the UV range where one of the wavelengths is highly absorbed by ozone, and is called the ``on-line'' wavelength. The other wavelength has a relatively lower absorption by ozone and is called the ``off-line'' wavelength.

As the ozone cross sections are well known, the differential lidar technique allows absolute number density to be determined from the combination of the on and off line measurements, without the need for external calibration.

*Comment: Page 3, line 10: The overlap function is defined but does not appear in the equation.*
Response: Thanks for catching this. We added it to Eq.1.

*Comment: Page3, lines 19-21: Quoting the absolute value of the absorption cross section of NO2 tells the reader nothing unless the cross section of ozone is also given.*
Response: the original sentence was not correct.
The original was:

The differential absorption cross section of $\mathrm {NO}_2$ in the specified spectrum is on the order of $3\times 10^{-19}$ $\mathrm{cm} ^{2}$, thus considering the effect of $\mathrm {NO}_2$ in the ozone retrievals is not essential.

We changed it to:

At mid-latitudes, the uncertainty of ozone number density due to absorption by $\mathrm {NO}_2$ reaches a maximum of 0.4\% between 25 and 30\,km altitude. Thus, the effect of $\mathrm {NO}_2$ on the ozone retrievals is not significant, and the third term of Eq.\ref{Optdepth} is small (Brasseur et al., 1999; Godin-Beekmann et al., 2003).

*Comment: Page 3, lines 23 & 26: Is "paralyzable" a real word?*
Response: Yes. A photomultiplier counting system  is called paralyzable when it fails to record an event (i.e measure photocounts) when the time interval between receiving two events is shorter than a given interval time, known as the deadtime l (\gamma). The Donovan at all reference cited in the paper is well know, and describes this effect in detail.

*Page 7, line 12: "The standard deviation to the 2 sigma level for this climatology is 50% below 25 km and 10%..." From the context I think you mean the standard deviation (i.e. 1-sigma), but this sounds like a confidence interval. In Table 1 some of the other parameters are described as standard deviations.*

Response: It is common in OEM to give variability of parameters as either 1 standard deviation (sigma; that is 67% of the variability is within this range) or 2 standard deviation (95% included). We will reword the sentence as:

The variability of the climatology we use is 50\% at the 2 sigma level, encompassing 95\% of the variability, and 10\% above 20\,km altitude.

*Comment: Page 7, lines 19, 30 (and elsewhere): "uncertainty" is not defined. Do you mean one standard deviation (1-sigma)?*

Response: We use uncertainty as a shorthand for "uncertainty of our knowledge of X", and perhaps one could argue variability is a better word to use, but we would like to leave it as uncertainty.

*Page 7, line 23 (and elsewhere): "correlation length" is not defined*

Response: We added the following sentence to the paragraph (third paragraph in section 3):

In the case of ozone and air density there is a vertical correlation between the elements of retrieval states. This corresponds to the off-diagonal elements of the \textit{a priori} covariance matrix. The correlation length gives the vertical correlation between the retrieval elements. It can be difficult to quantify the correlation length depending on the quantity. We have used a correlation length (\l$_{s}$) of 1000 \,m for ozone at altitudes below 18\,km and the correlation length of 1400\,m at higher altitudes. The air density has a correlation length of 1400\,m for all regions, which is about ⅕ of a scale height and consistent with vertical resolution of density measurements used for Rayleigh-scatter temperature lidar. It is beyond the range of this study, but feasible, that an extend ozonesonde record from a location could be used to better assess the correlation length for ozone density. The effect of using no correlation length would be to make the retrieval overly sensitive to measurement noise; using a very long correlation length would act to smooth the retrieval beyond the resolution of the retrieval grid. Neither extreme is the case here.

*Comment: Page 9, lines 1-2: An "uncertainty" of 19K for radiosondes sounds absurdly large. Uncertainties (1-sigma) for radiosondes are usually quoted in the range of 1K or less. I'm not familiar with MSIS uncertainties but 35K sounds pretty large. I can guess the temperature outside more accurately by simply looking at the calendar.*

Response: As discussed previously, in this use of uncertain in OEM it is meant to be uncertainty due to variability. In the paper, the uncertainty of temperature in sonde is written wrongly and is only about 1 K, we have corrected this in the paper. In the stratosphere and above Sica and Haefele (2015) showed a reasonable variability for

temperature is that given in the MSIS temperature model. Hedin et al. (1991) gives an uncertainty (as a RMS uncertainty) of 35 K over season and solar cycle. Hence, we adopt this uncertainty in the upper stratosphere and above.

HEDIN, A. E. (1991), Extension of the Msis Thermosphere Model Into the Middle and Lower Atmosphere, Journal of Geophysical Research-Atmospheres, 96(A2), 1159-1172.

*Comment: I think you mean "Retrieval a priori values"*
Response: Yes. It is corrected now.

*Comment:Page 10, lines 31-32: This description conflicts with that in the figure caption*
Response:
Original: The **red line** shows that the averaging kernel for ozone density equals 1 up to 42.7\,km, thus below this altitude the retrieval is independent of the \textit{a priori} profile.

We have changed the sentence to the following:
The **dashed line** shows that the area of the averaging kernel for ozone density equals 1 up to 42.7\,km. Below this altitude the retrieval is independent of the \textit{a priori} profile.

*Comment: Page 13, lines 20-24: Why is the lidar biased low in Figure 7? It seems to miss quite a bit of ozone.*
*Response:* The low bias is seen for the traditional ozone profile (the blue line), at lower altitudes in the traditional method. This bias is like due to saturation of the photomultiplier tube, despite including an empirical deadtime correction. For most nights the empirical correction is sufficient. Our OEM method retrieves the deadtime on a profile by profile basis for each digital detector channel.

*Comment: Page 13, line 25: Figure 8, not 7.*
Response: Thanks for catching this. It is fixed now.

*Comment: Captions, Figures 5 & 6: "...the maximum height at which the retrieval is independent from the a priori." is ambiguous (see Page 10, lines 31-32).*
Response: We changed all captions related to the dashed line to:

The horizontal dashed line is a height below which the OEM retrievals is more than 80% due to the measurements.

*Comment: Caption, Figure 7: "...horizontal dashed line shows the cut-off below which the effect of the a priori ozone profile is small less than 10%." We now have a third definition of this dashed line!*
Response: We fixed it.

*Comment: Figure 8 appears to be reversed (or the caption is wrong), as it shows the lidar higher than the sonde.*
Response:  Yes, it is (Sonde-OEM), we corrected this in our revised manuscript.

*Comment: Page 16, line 5: "The ozone retrieval extends from 12 km to 70.2 km." How much of the upper part is a useful measurement?*
Response: The ozone retrieval is useful to about  42.7 \km. We changed the sentence to:

The acceptable range of ozone retrieval extends from 12\,km to 43\,km...

---

## Author Response (AR1)

14 March 2019

Dear Dr. Hase:

We have uploaded detailed responses to both Referees. Also uploaded are 2 copies of the revised manuscript, one with colour highlighting to show removed and added materials.

Both Referees made many significant suggestions for the paper, all of which we accommodated. There were a few comments about the traditional ozone method used by OHP; where relevant we have included them in the revised manuscript, but we have fully discussed them in the responses (for instance there was a comment concerning coincident NASA lidar measurements at OHP, measurements which were not used in our manuscript, whose purpose is to introduce the advantages of using an Optimal Estimation Method to retrieve ozone profiles from a DIAL lidar, not evaluate the suitability of the traditional method).

We would like to thank the Referees for the time they spend helping us improve the manuscript.

Sincerely,
Bob Sica

[revised manuscript text omitted]